# A Transcriptomics-Based Bioinformatics Approach for Identification and In Vitro Screening of FDA-Approved Drugs for Repurposing against Dengue Virus-2

**DOI:** 10.3390/v14102150

**Published:** 2022-09-29

**Authors:** Madhura Punekar, Bhagyashri Kasabe, Poonam Patil, Mahadeo B. Kakade, Deepti Parashar, Kalichamy Alagarasu, Sarah Cherian

**Affiliations:** 1Dengue & Chikungunya Group, ICMR-National Institute of Virology, 20-A, Dr Ambedkar Road, Pune 411001, Maharashtra, India; 2Bioinformatics Group, ICMR-National Institute of Virology, 20-A, Dr Ambedkar Road, Pune 411001, Maharashtra, India

**Keywords:** repurposing drugs, DENV-2, dengue fever, antiviral therapy, FDA-approved drugs

## Abstract

The rising incidence of dengue virus (DENV) infections in the tropical and sub-tropical regions of the world emphasizes the need to identify effective therapeutic drugs against the disease. Repurposing of drugs has emerged as a novel concept to combat pathogens. In this study, we employed a transcriptomics-based bioinformatics approach for drug identification against DENV. Gene expression omnibus datasets from patients with different grades of dengue disease severity and healthy controls were used to identify differentially expressed genes in dengue cases, which were then applied to the query tool of Connectivity Map to identify the inverse gene–disease–drug relationship. A total of sixteen identified drugs were investigated for their prophylactic, virucidal, and therapeutic effects against DENV. Focus-forming unit assay and quantitative RT-PCR were used to evaluate the antiviral activity. Results revealed that five compounds, viz., resveratrol, doxorubicin, lomibuvir, elvitegravir, and enalaprilat, have significant anti-DENV activity. Further, molecular docking studies showed that these drugs can interact with a variety of protein targets of DENV, including the glycoprotein, the NS5 RdRp, NS2B-NS3 protease, and NS5 methyltransferase The in vitro and in silico results, therefore, reveal that these drugs have the ability to decrease DENV-2 production, suggesting that these drugs or their derivatives could be attempted as therapeutic agents against DENV infections.

## 1. Introduction

Dengue fever (DF) and dengue haemorrhagic fever are mosquito-borne diseases caused by the dengue virus serotype (DENV-1, 2, 3, and 4) [1]. The incidence of DENV has grown dramatically around the world in recent decades. The World Health Organization (WHO) estimates 390 million dengue infections per year in 128 countries [2]. In the tropics and subtropics, dengue is one of the most common diseases, which is a public health concern. DENV, a positive-strand RNA virus, is a member of the family Flaviviridae and genus flavivirus, transmitted by *Aedes aegypti* and *Ae. albopictus* mosquitoes [3]. The DENV infection usually causes dengue fever (DF) with flu-like illness, and DF occasionally evolves into a potentially lethal complication termed severe dengue (SD), which is characterized by plasma leakage, fluid accumulation, respiratory distress, and severe bleeding [4].

To date, there is no treatment or approved antivirals available for DENV. The only available methods are mosquito-control strategies which prevent the transmission of the virus, and hence, there is an urgent need to design better medications for treating dengue viral infection. The traditional drug discovery procedure is more expensive, can take many years, and requires unending work to find a novel drug. This problem can be overcome by drug repurposing or drug repositioning, which involves a drug discovery strategy for the United States Food and Drug Administration (FDA)-approved drugs [5]. Among the many approaches employed to identify drugs to treat dengue, drug repurposing has gained popularity, which is safer and more economic. The targeted drugs have further been tested for their effectiveness against other diseases and proven safe for treatment of the human diseases [5]. In recent years, drug repurposing has been applied in many studies to identify treatments [6,7,8]. Antidiabetic, anticholesteremic, antihistamine, antipsychotic, antibiotic, antiparasitic, and antimalarial drugs have been repurposed to assess their effectiveness against DENV infection [5,9]. These studies helped uncover potential drug targets for improving DENV therapy.

In this study, we applied computational drug-repurposing methods by using publicly available gene expression data for gene–disease–drug analysis. The differentially expressed genes were inputted to the connectivity map (CMap) server to identify drug candidates for DENV. The identified candidate drugs were tested in vitro for anti-DENV activity. The drugs with antiviral activity were further investigated for their ability to interact with DENV proteins using molecular docking analysis.

## 2. Materials and Methods

The schematic representation of the in silico as well as in vitro study workflow is shown in Figure 1.

### 2.1. Transcriptome Data Collection and Analysis

Identification of differential gene expression between dengue patients and healthy individuals.

Several transcriptomic studies (microarray and RNASeq) were referred for the expression analysis of dengue virus infection in host cells. Three microarray gene expression datasets based on whole blood samples of dengue-infected humans were downloaded from the GEO database as follows: GSE18090 [10] from blood samples of eight healthy and ten DHF patients, GSE51808 [11] from nine healthy and ten DHF patients, and GSE84331 [12] from five healthy and three DHF patients. For each dataset, the gene signature profiles were obtained by considering an adjusted *p* (Adj. *p*)-value < 0.05 in the test between healthy and DHF patients by the z-score transformation of the three GEO matrices.

### 2.2. Identification of Drug Candidates Using CMap

CMap is a collection of genome-wide transcriptional expression data from human cell lines that have been treated with chemical compounds [13]. Simple pattern matching algorithms together enable the discovery of decisive functional connections between drugs, genes, and diseases through the transitory feature of common gene expression changes. The CMap has a library containing over 1.5 million gene expression profiles from ~5000 small-molecule compounds and ~3000 genetic reagents, tested in multiple cell types. The query tool on the clue.io site helps in identifying perturbens that give rise to similar (or opposing) gene signatures [14]. In drug repositioning for dengue, the differentially expressed signature profiles between healthy individuals and DHF patients were applied to the query tool to identify inverse drug–disease relationships. The threshold of significance for each drug was set at *p* < 0.1 using the permutated results. Once queried, the results were accessible, and the heat map result of the connections was viewed. In the heat map, the liver cell line was selected as HEPG2 and the perturben type was selected as a compound. Thereafter the potential therapeutic candidates were identified among the compounds with the highest (negative) connectivity score. Connectivity scores less than −90 were considered significant according to Connectopedia, the Clue knowledge database.

### 2.3. Cells and Virus

The Vero CCL-81 (ATCC^®^ CCL 81™) cell line, originally derived from the African green monkey kidney, was used to assess antiviral activity. These cells were cultured at 37 °C and under 5% CO_2_ in Minimal Essential Medium (MEM), (HiMedia^®^, Mumbai, India) with 10% foetal bovine serum (FBS) (Gibco^TM^, Grand Island, NY, USA) and antimycotic antibiotic solution (Sigma Aldrich^®^, St. Louis, MO, USA). The DENV-2 virus (Strain No. 803347) was used in this study and a 0.1 Multiplicity of Infection (MOI) was used for infection.

### 2.4. Stock Preparation of Compounds

The drugs procured from Sigma Aldrich, St. Louis, MO, USA, (Table 1) were included in this study. The 20 mM stock solutions of the drugs were prepared using dimethyl sulfoxide (DMSO, 100%) as a diluent. The stock solutions were stored at −20 °C for further analysis.

### 2.5. Cytotoxicity Screening of Compounds

The cytotoxicity screening of the drugs was performed using a 3-(4,5-dimethythiazol-2-yl)-2,5-diphenyl tetrazolium bromide (MTT) assay as described in a previous study [71]. The Vero CCL-81 cells cultured in a 96-microtitre-well plate with a density of 2 × 10^4^ cells per well were used with 80% confluency. The compound concentrations were prepared using MEM without antibiotics and with 2% foetal bovine serum (FBS) as the diluent, and a twofold dilution technique (200 to 0.78 µM) was adopted.

The cells were then incubated with these drugs in different microtiter plates for five days (120 h) at 37 °C with 5% CO_2_, and further incubated with MTT solution (5 mg/mL) for three hours at 37 °C. A microplate reader (BioTek Synergy, Santa Clara, CA, USA) was used to measure the solubilized formazan crystals at 570 nm with a reference filter of 690 nm. Further, we computed the percentage viability of the cells at different concentrations of the drug in comparison with cell control, and CC_50_ values of the drugs were calculated using non-linear regression analysis using GraphPad Prism software version 7.

### 2.6. Antiviral Assay

For screening, the maximum non-toxic concentration of the drugs was selected for the assessment of antiviral activity before infection (pre-treatment), during infection (co-treatment), and after infection (post-treatment). For all three treatments, 5 × 10^4^ cells/well were used with 0.1 multiplicity of infection (MOI) of DENV-2. After infection, the cells were washed twice to remove unbound virus particles and the plates were incubated for five days as per the treatment condition.

Pre-treatment condition involved incubation of the drug onto the cells for 24 h prior to infection. For post-treatment, the cells were infected with the virus for one hour and treated with the drug for five days. In the case of co-treatment, the virus and drug mixture were incubated for one hour and later this mixture was added to the cells and incubated for one hour, followed by the washing step and incubation of five days.

The viral genomic RNA and infectious virus particle titre of cell culture supernatants taken from separate wells treated with different concentrations were analysed using real-time RT-PCR and focus-forming unit (FFU) assay.

Drugs which showed anti-DENV activity were again tested for their anti-DENV activity at different concentrations ≤ maximum non-toxic dose to find out the minimal dose that exerts antiviral activity. The same treatment condition under which the drug showed activity was used for these dose-dependent studies. A virus control (VC) in which cells were infected but not treated was also maintained. The experiments were performed in triplicate at three independent time points.

### 2.7. Focus Forming Unit Assay and Real-Time RT-PCR

For the quantification of virus particles, FFU assay was used. The assay was performed in a 96-well plate as described earlier [72,73]. Approximately 2 × 10^4^ cells/well were seeded in a 96-well plate and incubated for 24 h. Tenfold serial dilutions of the culture supernatants were added to the cells and the plates were incubated for one hour. After incubation, MEM with 2% FBS and 1.8% carboxymethyl cellulose was added and incubated at 37 °C for five days in a CO_2_ incubator. After incubation, cells were washed with phosphate-buffered saline (PBS) containing Tween 20 detergent and fixed with chilled acetone and methanol (1:1 ratio). A blocking buffer (1% bovine serum albumin dissolved in PBS) was added and incubated for 40 min at 37 °C. Later, the washing step was again performed, and the cells were incubated with primary antibody (anti-prM-dengue antibody with a dilution of 1:200) for 40 min, followed by the addition of secondary antibody (anti-mouse IgG HRP conjugate with a dilution of 1:1000) and incubated for 40 min. After every addition, a washing step was performed. The last step involved the addition of substrate (True Blue Peroxidase Substrate by KPL) in dark, followed by incubation at room temperature for 15 min. The substrate was removed after the blue tinge formation and dried before counting the number of foci. The virus titre was determined by counting the number of foci.

Quantitative real-time RT-PCR (qRT-PCR) for estimating the copy number of DENV-2 was performed as described earlier [72,73]. A QIAmp Viral RNA mini kit (Qiagen, Hilden, Germany) was used for the extraction of viral RNA. One-step qRT-PCR using a commercial kit (SuperScript III One-Step RT-PCR System with Platinum Taq DNA Polymerase; ThermoFisher Scientific, Waltham, MA, USA) (Invitrogen SuperScript III Platinum One-step qRT-PCR Kit, Waltham, MA, USA) was used for the detection and quantification of RNA. Oligonucleotide sequences were used as described earlier [72,73]. The amplification conditions included incubation at 50 °C for 30 min, followed by 95 °C for 10 min, and 40 cycles of 95 °C for 15 s and 60 °C for one minute. The copy numbers were calculated based on a standard graph generated using Ct values of tenfold dilutions of in vitro-transcribed viral RNA with known copy numbers. The assays were repeated for the compounds which showed inhibitory activity compared to the virus control.

### 2.8. Statistical Analysis

The virus output was expressed as log_10_ mean viral RNA copies/mL or log_10_ mean FFU/mL. One-way ANOVA with correction for multiple comparisons was used to compare the test conditions to the VC. A *p*-value less than 0.05 was considered significant. All analysis was performed using GraphPad Prism software version 7 (GraphPad software, San Diego, CA, USA).

### 2.9. Molecular Docking Studies with DENV Protein Targets

The Schrödinger Drug Discovery Suite was used for protein preparation and docking calculations (2020-3). The two-dimensional structures of all the compounds were retrieved from PubChem [74]. All DENV-2 three-dimensional (3D) crystallographic target protein structures including envelope glycoprotein (1OKE.pdb), DENV-2 non-structural proteins including NS2B-NS3 protease (4M9K.pdb), NS5 methyltransferase (MTase) domain (1R6A.pdb), and NS5 RdRp (RNA dependent RNA Polymerase) domain structure (5ZQK.pdb) were retrieved from the Protein Data Bank (PDB) [75] (https://www.rcsb.org/ (accessed on 15 June 2022). The protein structure was prepared (pre-processed, optimized, and minimized) using the protein preparation wizard in Maestro v. 12.5 [76]. The compound structures were prepared by the LigPrep module of Maestro v. 12.5 [76,77]. Grid-based molecular docking was used to help the compounds bind in multiple potential conformations. The hydronation and tautomeric modes of the amino acids were adjusted to match a pH of 7.2. The receptor grid-generating module of Maestro v.12.5 was used to generate the grid. A scaling factor of 1.0, and 0.25 Å, partial charge cut-off for van der Waals radius were applied.

The detergent binding pocket (β-D-glucoside β-OG) of the envelope protein was used as the active site [78] as this pocket is identified as a potential site for small-molecule fusion inhibitors against the envelope protein. Further, the allosteric site pocket for NS2B-NS3 protease [79], conserved motifs, and priming loop for NS5 RdRp domain [80], as well as the site map-generated pocket that was used for NS5 methyltransferase domain, were selected for the molecular docking protocol. The drug binding site was predicted through the site map module of Maestro v.12.5 (Schrödinger, New York, NY, USA).

The docking was carried out using SP/XP (Standard/extra precision) mode [76,77]. The study of DENV-2 protein binding interactions and ligand flexibility was carried out using Glide molecular docking. The binding energies were measured in kcal/mol. The BIOVIA Discovery Studio 2020 client software suite was used to analyse interactions and visualise molecules of docked complexes of all the DENV target crystal structures. The compounds with the highest binding scores and strong interaction profiles were supposed to be the most active compounds against the target receptor protein. The crystal structure was later refined and optimized for further docking procedures.

## 3. Results

### 3.1. Transcriptomic Analysis

Using the GSE18090 dataset, we identified 31 signature genes (4 upregulated and 27 downregulated) with significant differences in expression between the DHF patients and the 8 healthy controls. Additionally, we identified 3858 (148 upregulated and 3710 downregulated) and 1941 (541 upregulated and 1400 downregulated) differentially expressed genes in the GSE51808 and GSE84331 datasets, respectively. The genes were filtered out based on the cut-off criteria of adj. *p*-value < 0.015 and log fold change >2. After combining the signature genes from all three GEO datasets, 1585 signature genes were obtained (Appendix A).

### 3.2. CMap Analysis

The list of the top 300 upregulated/downregulated signature genes from the three datasets that were compatible with the HG-U133A platform of CMap, was used to query the clue.io CMap system. The valid list of differentially expressed genes from the three datasets was queried on CMap for the drug candidates and hits with significant inverse gene signatures (connectivity score < −90) were shortlisted. There were a total of 200 significant drugs (Appendix A) obtained with statistical significance (*p* < 0.1). This includes different groups such as CDK inhibitors, JAK inhibitors, interferons, angiogenesis inhibitors, and HDAC inhibitors (Figure 2).

### 3.3. Viral Targeting Drugs Identified for Repurposing against DENV

Among the list of drug compounds identified by CMap analysis, some are direct viral targeting such as flavivirus RNA polymerase inhibitor, protease inhibitor, etc., while others are host targeting such as histone deacetylase (HDAC) inhibitor. In this study, 16 of these drug compounds were tested for their antiviral efficacy against DENV-2 and evaluated. The profile of these compounds is listed in Table 1.

### 3.4. Cytotoxicity Evaluation of Compounds

The cell viability of all the drugs (*n* = 16) was tested using an MTT assay in Vero CCL-81 cells. The CC_50_ values of all the drugs are provided in Table 1. Seven drugs were highly toxic. The concentration of the drug which allowed approximately 80% cell viability was further used for studying the antiviral activity. In Appendix A, the impact of different drugs at various concentrations on cell viability is shown with CC_50_ values.

### 3.5. Primary Antiviral Screening of Compounds against DENV-2

The drugs were tested for their antiviral activity under three circumstances (Figure 3a–c), i.e., pre-treatment (prophylactic), co-treatment (virucidal), and post-treatment (therapeutic). Focus-forming unit (FFU) assay was used to quantify the virus particles. Out of 16 compounds, five compounds (doxorubicin, resveratrol, lomibuvir, elvitegravir, and enalaprilat), showed antiviral activity (≥1.0 log10 reduction compared with virus control) against DENV-2 under different treatment settings. Three drugs, i.e., resveratrol and lomibuvir, had both prophylactic and therapeutic antiviral activity against DENV-2. Elvitegravir had only prophylactic antiviral activity against DENV-2. Doxorubicin had both virucidal and therapeutic antiviral activity against DENV-2. Enalaprilat had only therapeutic antiviral activity against DENV-2 (Table 2). As a result, these drugs were selected for further exploration in a dose-dependent manner, as well as in in silico experiments to determine their target specificity.

### 3.6. Dose-Dependent Antiviral Effect of Repurposed Drugs against DENV

Dose-dependent antiviral effect of effective drugs, i.e., doxorubicin, resveratrol, lomibuvir, elvitegravir, and enalaprilat, on DENV-2 was examined under different treatment settings and concentrations. The effective drugs were categorised under three treatment settings: prophylactic, therapeutic, and virucidal.

#### 3.6.1. Drugs Exerting Prophylactic Effects

Resveratrol: The pre-treatment of cells with 12.5 µM resveratrol showed >1 log_10_ reduction (*p* < 0.0001) in the virus titre compared to VC (5.445 to 4.257 mean log_10_ FFU/mL) (Figure 4). Resveratrol treatment did not result in a significant reduction in the log_10_ titre of viral RNA copy number (Appendix A).

Lomibuvir: Under pre-treatment conditions, lomibuvir exerted one log reduction in virus titre (5.445 to 4.221 mean log_10_ FFU/mL) at 6.25 µM concentration (Figure 4). Quantitative real-time RT-PCR results revealed that the drug did not affect viral RNA titre under pre-treatment condition (Appendix A).

Elvitegravir: This drug at 6.25 µM concentration of the drug showed a reduction from 5.445 (VC) to 4.129 mean log_10_ FFU/mL (Figure 4) but there was no reduction in the case of viral RNA titre (Appendix A).

#### 3.6.2. Drugs Exerting Therapeutic Effects

Doxorubicin: A substantial reduction in virus foci (5.424 to 3.971 mean log_10_ FFU/mL value) was detected in cells which received post infection treatment with a 25 µM concentration of drug (*p* < 0.0001) (Figure 5). The drug did not result in a significant reduction in the log_10_ titre of viral RNA copy number under post-treatment condition (Appendix A).

Resveratrol: Resveratrol at 12.5 µM and 6.25 µM concentrations showed ≥1 log_10_ reduction (5.424 to 4.198 mean log_10_ FFU/mL) (*p* < 0.0001) of the virus titre compared to VC (Figure 5). There was no significant reduction observed in the log_10_ titre of viral RNA copy number (Appendix A).

Enalaprilat: This drug showed more than one log reduction in the case of post-treatment (5.424 to 4.184 mean log_10_ FFU/mL) at 1.56µM concentration (Figure 5). However, the post-treatment condition did not show a corresponding reduction in viral RNA titre. (Appendix A).

#### 3.6.3. Drugs Exerting Virucidal Effects

Doxorubicin: A detailed study was performed with different concentrations (25 µM, 12.5 µM, 6.25 µM, and 3.125 µM) of doxorubicin and the co-treatment of cells with 25 µM concentration showed ~100% reduction (~5 log_10_) (*p* < 0.0001); at 12.5 µM (*p* < 0.0001) and 6.25 µM, the virus titre showed a difference of ~2 log_10_FFU titre (*p* < 0.0001) when compared with VC (Figure 6). A significant 4 log_10_ titre (*p* < 0.0001) decrease in RNA copy number of DENV-2 was observed for co-treatment at 25 µM, whereas a ~3 log_10_ reduction was observed at 12.5 µM (*p* < 0.0001) and 6.25 µM (*p* < 0.0001) concentrations (Appendix A).

### 3.7. Effect of Different Drugs on the Percent Cell Viability of the Infected Cells

To rule out that the anti-DENV activity exerted by different drugs was not due to cell death, the cells were cultured in 96-well plates, infected, and treated with a maximum non-toxic dose of the drugs which exerted anti-DENV activity under different treatment conditions. In one set of experiments, cells were incubated for three days after incubation while in another set cells were incubated for five days. After incubation, an MTT assay was performed and the percent cell viability in each well was calculated using viability in cell control (uninfected and untreated cells) as reference. The results revealed that irrespective of days and the treatment condition, the percent cell viability was greater for infected cells which received treatment compared with infected cells which did not receive treatment (virus control) (Figure 7a–c). The only exception was enalaprilat for which the percent cell viability was not different compared to VC (Figure 7b). This proved that the increase in percent cell viability was due to drug treatment which inhibited the virus replication and subsequent cytopathic effect.

### 3.8. In Silico Interaction Studies of the Select Drugs with DENV Protein Targets

To explore the potential mode of action of the drugs resveratrol, doxorubicin, lomibuvir, elvitegravir, and enalaprilat, computational molecular docking studies were performed with DENV targets using Schrödinger’s Glide. Following the docking procedure, the best pose was chosen based on conformation and docking energy. The scoring function algorithm implemented in Schrodinger was used to compute the binding affinity in terms of the docking score.

Among the different DENV targets, the docking interaction of doxorubicin was found to be the best with DENV E protein (Figure 8e) which exhibited that doxorubicin docked with a strong binding affinity of −5.8 kcal/mol. The doxorubicin interaction involves two conventional hydrogen bonds, four pi-alkyl, and a single pi-sulphur interaction. The residues THR 236, PRO 217, ALA 263, and MET 260 were involved in the above interactions.

The docking interaction analysis of resveratrol was found to be optimal with the DENV NS5 MTase domain (Figure 8a). The binding affinity was found to be −5.7 kcal/mol and residues ARG78, LYS99, and ASP 140 made three hydrogen bonds, indicating that resveratrol can make a stable complex with the DENV MTase domain.

The DENV NS2B-NS3 protease is one of the most prominent targets for anti-DENV inhibitors. Aside from the main catalytic location (HIS51, ASP75, and SER135), a specific allosteric binding site is also present, which has been previously suggested for non-competitive inhibitors. The enalaprilat docking results (Figure 8b) showed that it interacted with allosteric site residues of DENV NS2B-NS3 protease with a binding affinity of −3.02 kcal/mol. A total of seven interactions were formed: four hydrogen bonds formed with allosteric site residues VAL1155, ASN1119, ARG1157, and LYS1117 and three hydrophobic interactions formed with VAL1155, ILE1123 and VAL1154. These interactions reveal that enalaprilat could have inhibiting activity at the allosteric site of NS2B-NS3 protease.

Examining the docking contact of elvitegravir with the NS5 RdRp domain (Figure 8c) highlighted a binding affinity of −5.27 kcal/mol. The potential binding site of elvitegravir was close to the catalytic site showing interactions with the residues of all the three conserved motifs (Q598-N614, G662-D664, and C709-R729) as well as residues of the priming loop (H786-M809). The molecular interactions displayed two hydrogen bonds formed by ARG729 and the hydroxyl group of elvitegravir. Also, three other hydrogen bonds were formed with THR794, THR 793, and LYS460. A single pi-cation interaction was noted with ARG729. There were multiple numbers of hydrophobic interactions, including alkyl and pi-alkyl bonds with ILE797, TRP795, and SER710.

The docking interaction study of lomibuvir with the DENV RdRp domain (Figure 8d) revealed a strong binding affinity of −4.9 kcal/mol. The molecular non-bonded interaction analysis displays that there were three hydrogen bonds with LYS195, GLN337, and ASP 397, two alkyl bonds with ILE531, VAL338, one salt bridge with LYS195, and one electrostatic bond with LYS 195.

## 4. Discussion

Dengue fever has been a serious public health problem in tropical and sub-tropical areas around the world and the only preventive measure available for the dengue virus is the control of mosquitoes. The development of effective and safe dengue virus vaccines is still a challenging topic because of antibody-dependent enhancement (ADE) [81]. ADE accounts for the severity of secondary infections that develop in the form of dengue haemorrhagic fever (DHF) and dengue shock syndrome [82]. Among the four serotypes of DENV, DENV-2 significantly causes more severity in patients than the other three serotypes [83]. Since there is no cure for DENV, repurposing known compounds could be a life-saving consequence for the majority of the world’s population. Several studies report anti-DENV activity using repurposed compounds [5,84,85,86,87].

In this study, we incorporated different classes of drugs including antineoplastic, anti-inflammatory, antibiotics, adrenal steroid synthesis inhibitors, antiviral, steroids, neuromuscular blockers, calcium channel blockers, enzyme inhibitors, hydroxamate inhibitors, and bromodomain inhibitors as shortlisted from a transcriptomics-based approach.

Among the five drugs (resveratrol, doxorubicin, lomibuvir, elvitegravir, and enalaprilat) which showed antiviral activity, doxorubicin, resveratrol, and enalaprilat were the only drugs which showed maximum inhibition of DENV-2 in vitro under the post-treatment condition. Resveratrol is a naturally occurring polyphenol, which is known to exhibit anti-inflammatory, antineoplastic, anti-diabetic, antilipemic, antioxidant, antimicrobial, as well as antiviral activity [41,42,43,44,46,47,85]. The other activities include a cardioprotective mechanism and a modulator of Alzheimer’s disease [45,48]. The docking analysis revealed that resveratrol interacted with the DENV NS5 MTase domain and formed a stable complex with it and might account for the antiviral activity of the drug. An earlier study reported that resveratrol inhibited DENV-2 in Huh-7 cells but at a higher concentration in the range of 50–100 µM [88]. Resveratrol has been shown to inhibit DENV-2 infection in HEK293T/17 cells at concentrations from 25 to 50 µM [88]. In the present study, we have demonstrated the anti-DENV activity at lower concentrations (in the range of 6.25–12.5 µM). Resveratrol also inhibits the translocation of high mobility group box 1 protein from the nucleus to the cytoplasm and also increased the expression of interferon-stimulated genes in DENV-infected Huh-7 cells [88]. Thus, resveratrol might confer an antiviral state and may act as a prophylactic as well as a therapeutic agent.

Doxorubicin, an anthracycline antibiotic, isolated from *Streptomyces peucetius var. caesius*, exhibits antineoplastic activity by intercalating between DNA base pairs [19]. Many studies have reported that doxorubicin can inhibit a variety of viruses including herpes simplex virus, HIV, enterovirus, hepatitis C virus (HCV), and SARS-CoV-2 [20,21,22,23]. An in vitro study by Kaptein et al. reported that doxorubicin and its derivatives inhibit the flavivirus replication cycle [24]. In the present study, doxorubicin exerted both virucidal as well as a therapeutic effect against DENV-2. In silico studies revealed that doxorubicin may interact with the E protein and binds specifically to the hydrophobic pocket in which residues of the fusion peptide are also accommodated [73]. It is conceivable that doxorubicin may interact with the DENV envelope protein preventing its binding to the host cell receptor. The main limitation posed by doxorubicin concerns the reports of its cardiotoxicity induced by long-term doxorubicin exposure [89]. The structure of doxorubicin might provide clues to the synthesis of derivatives and molecules which can bind to DENV E protein with high affinity but with no cardiotoxic effects.

Enalaprilat, an angiotensin-converting enzyme (ACE) inhibitor, is an active metabolite of enalapril prodrug and is used for the treatment of hypertension, heart failure, and chronic renal diseases. Enalaprilat is known to inhibit the conversion of angiotensin I to angiotensin II by acting antagonistically against the angiotensin-converting enzyme (ACE) [90]. Enalaprilat exerted therapeutic antiviral activity against DENV-2. Enalaprilat interacted with DENV NS2B-NS3 protease at the proposed allosteric binding site for non-competitive inhibition, though with a low binding affinity. It is possible that enalaprilat can affect the DENV protease activity and contribute to the inhibition of DENV. Angiotensin II inhibitors, by acting either on the angiotensin II receptor or ACE, have been shown to inhibit DENV and Zika virus [91,92]. Enalaprilat might modulate the host response and contribute to virus inhibition.

Lomibuvir is a non-nucleoside polymerase inhibitor, belonging to the thiophene carboxylic acid group. It is used in the treatment of chronic HCV infection, and the mechanism involves binding to the thumb pocket 2 of the HCV NS5B polymerase [68,69]. Elvitegravir was the first HIV-1 inhibitor approved by the Food and Drug Administration (FDA) in 2012. It acts as an integrase strand transfer inhibitor for HIV-1 [66,67]. In silico studies revealed that both drugs interacted with the DENV NS5 RdRp domain. However, the in silico results are not supported by in vitro results since both lomibuvir and elvitegravir showed prophylactic activity against DENV infection but no therapeutic activity. It is possible that both these drugs can bind to host receptors and prevent binding of DENV or might induce an antiviral state contributing to inhibition of DENV infection. Further studies are needed to confirm the same.

Overall, in this study, we have identified five potential drugs that can be repurposed as DENV therapeutics. In vivo validation studies would further help in consolidating the findings and facilitate clinical trials.

## Figures and Tables

**Figure 1 viruses-14-02150-f001:**
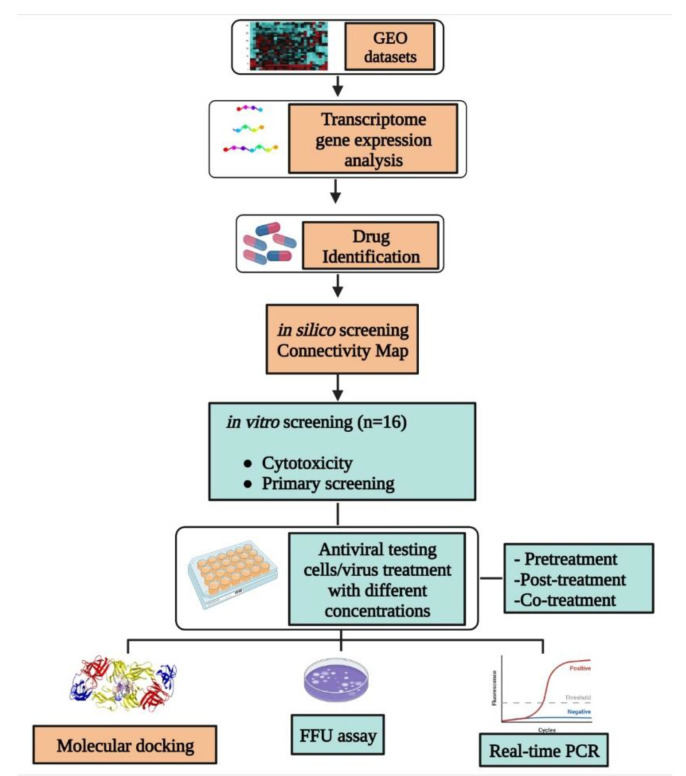
In silico and in vitro methods used for selecting and studying the antiviral activity of drugs for repurposing.

**Figure 2 viruses-14-02150-f002:**
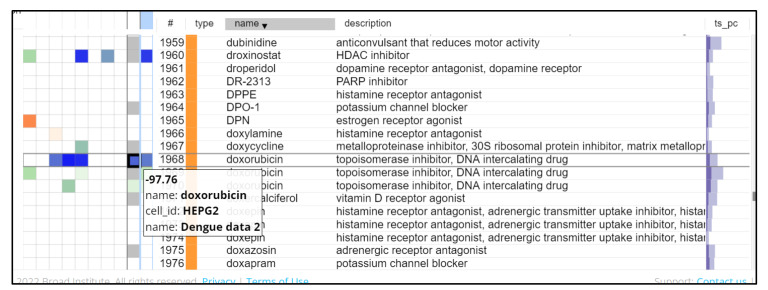
Snapshot of the heat map generated by querying the DHF DEGs in CMap. The blue coloured gradient indicates the negative connectivity score of the compounds with inverse gene signatures.

**Figure 3 viruses-14-02150-f003:**
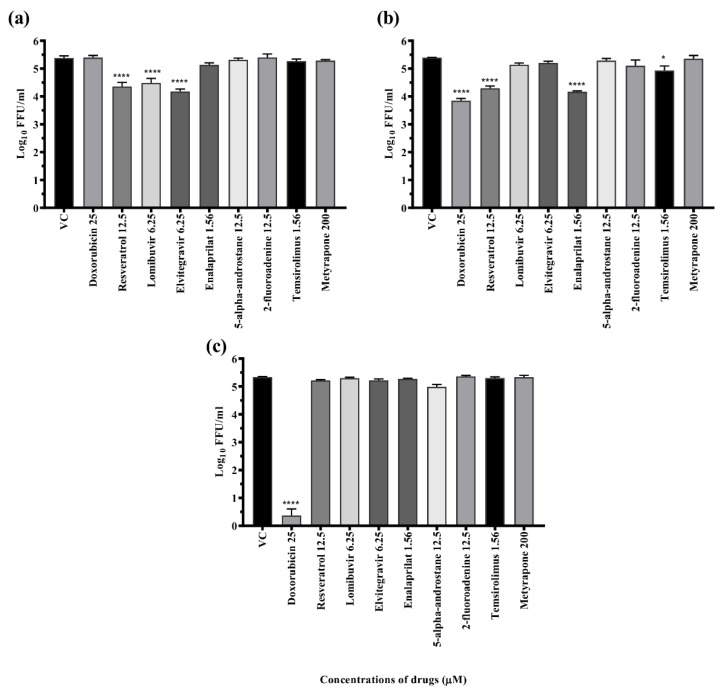
Antiviral screening of drugs for repurposing at maximal nontoxic concentration against DENV under (**a**) pre-, (**b**) post-, and (**c**) co-treatment conditions. Vero CCL-81 cells were treated with a maximum non-toxic dose of drugs for 24 h in a pre-, co-, and post-infection manner and incubated for 120 h with DENV. Post incubation, plates were frozen and culture filtrates were used for different assays. Experiments were performed in triplicate in two independent trials. Results were plotted based on mean log10 of focus-forming unit/mL ± standard error. All the treatment groups were compared with a control group (VC) (infected cells without treatment). **** *p* < 0.0001, * *p* < 0.05.

**Figure 4 viruses-14-02150-f004:**
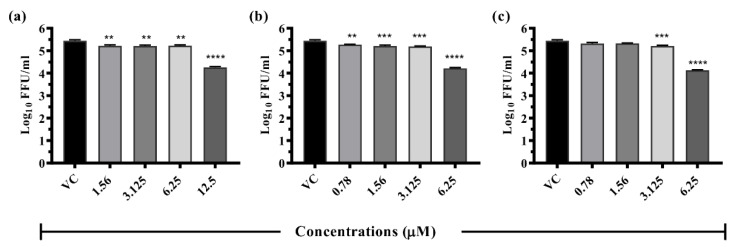
Antiviral effects of resveratrol, lomibuvir, and elvitegravir against DENV under pre-treatment condition. Vero CCL-81 cells were treated with different concentrations of respective drugs for 24 hrs and infected with DENV-2 and incubated for 120 h. The cultured filtrates were used for the FFU assay: (**a**)—resveratrol, (**b**)—lomibuvir, and (**c**)—elvitegravir. Experiments were performed in triplicate at three independent time points. Results were plotted based on mean log_10_ of focus-forming unit/mL ± standard error. All the treatment groups were compared with a virus control (VC) group (infected cells which did not receive drugs). **** *p* < 0.0001, *** *p* < 0.0005, ** *p* < 0.01.

**Figure 5 viruses-14-02150-f005:**
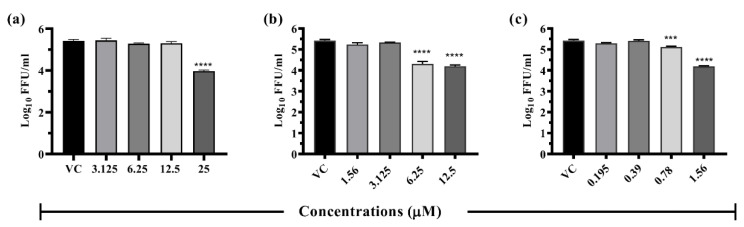
Antiviral effects of doxorubicin, resveratrol, and enalaprilat against DENV under post-treatment conditions. Vero CCL-81 cells were administered the drug at different concentrations immediately after infection and incubated for 120 h after infection. The cultured filtrates were used for the FFU assay: (**a**)—doxorubicin, (**b**)—resveratrol, and (**c**)—enalaprilat. Experiments were performed in triplicate at three independent time points. Results were plotted based on mean log10 of focus-forming unit/mL ± standard error. All the treatment groups were compared with a virus control (VC) group (infected cells which did not receive drugs). **** *p* < 0.0001, *** *p* < 0.0005.

**Figure 6 viruses-14-02150-f006:**
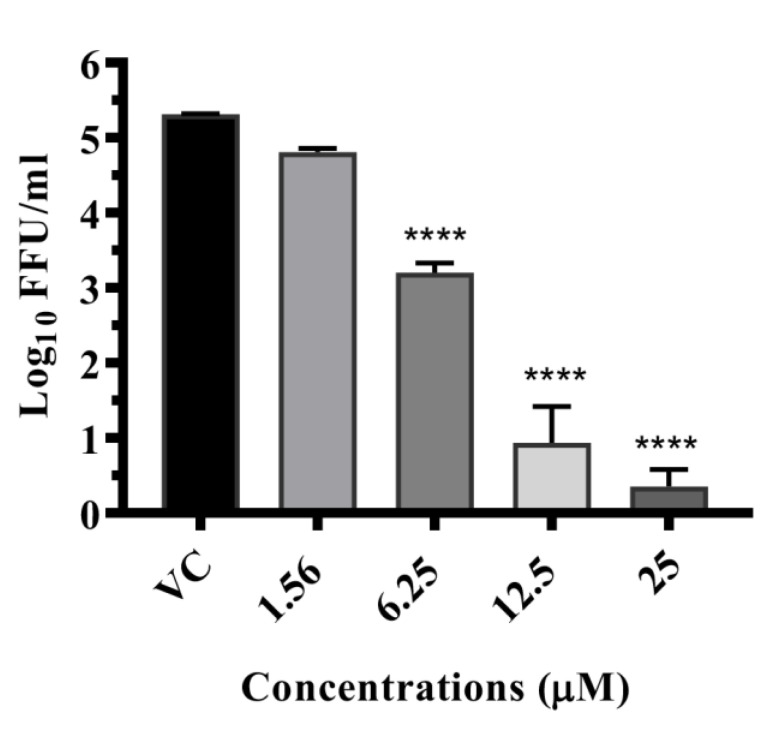
Antiviral effects of doxorubicin under co-treatment condition. The virus was treated with doxorubicin at different concentrations and the virus drug mixture was used for infection. Infected Vero CCL-81 cells were incubated for 120 h after infection. The culture filtrates were used for the FFU assay. Experiments were performed in triplicate at three independent time points. Results were plotted as mean log10 of focus-forming unit/mL ± standard error. All the treatment groups were compared with a virus control (VC) group (infected cells which did not receive drugs). **** *p* < 0.0001.

**Figure 7 viruses-14-02150-f007:**
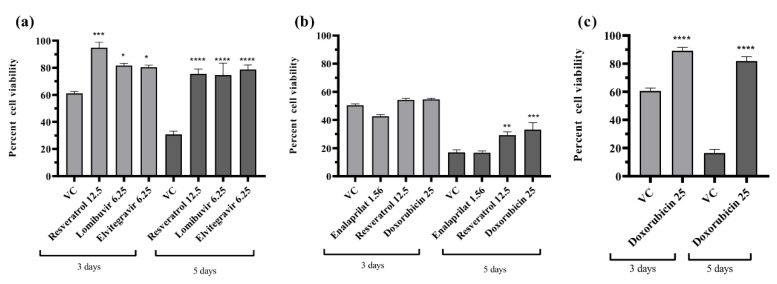
Effect of various drugs on the percent cell viability of the infected cells under different treatment conditions. (**a**)—pre-treatment, (**b**)—post-treatment, and (**c**)—co-treatment. **** *p* < 0.0001, *** *p* ≤ 0.0005, ** *p* < 0.01, * *p* < 0.05.

**Figure 8 viruses-14-02150-f008:**
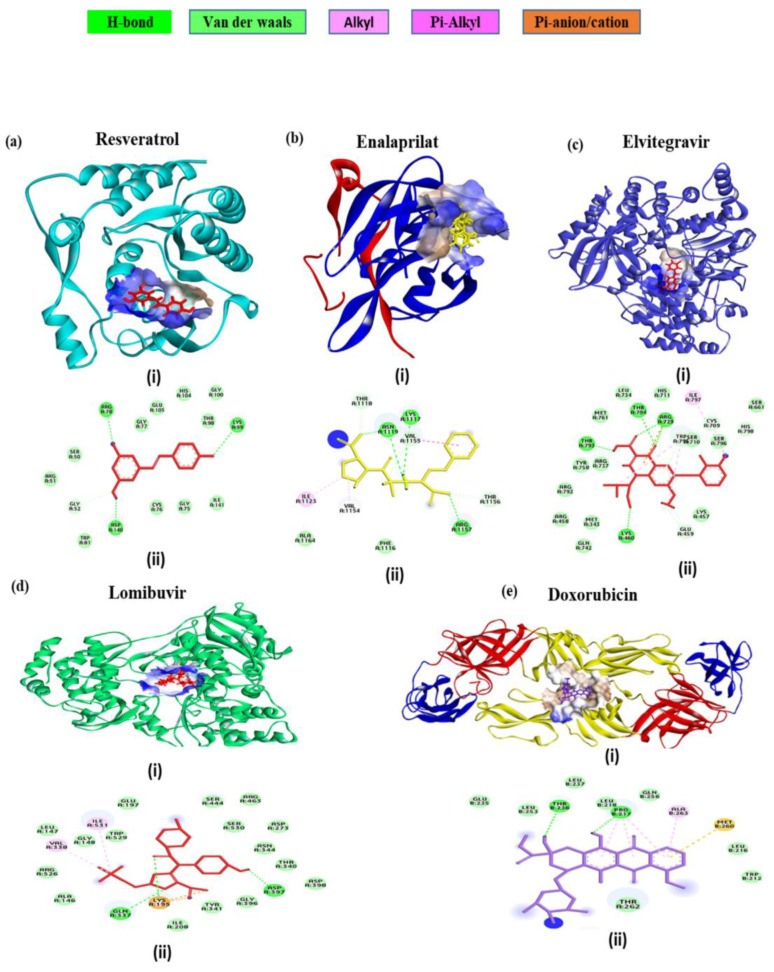
Molecular docking interaction of five repurposed drugs with DENV structural and non-structural proteins. Ribbon diagram with the solvent surface rendered view (probe radius 1.4 Å) and 2-dimensional interaction diagram showing interaction with DENV: (**a**) NS5 methyltransferase domain, (**b**) NS2B-NS3 protease domain, (**c**) NS5 RdRp domain, (**d**) NS5 RdRp domain, and (**e**) Envelope glycoprotein complex. (i)—3D interaction diagram, (ii)—2D interaction diagram.

**Table 1 viruses-14-02150-t001:** Profile of the drugs repurposed as DENV-2 inhibitors.

Sr. No.	Compound Name	Pharmacological Class	CC_50_ Value (µM)	Reported Activity
1.	Temsirolimus	Antineoplastic	12.24	1. Antineoplastic [15,16] 2. Antiviral, anti-SARS-CoV-2, and anti-HBV [17,18]
2.	Doxorubicin hydrochloride	Antineoplastic	116.9	1. Antineoplastic [19] 2. Antiviral [20,21,22,23,24]
3.	2-Fluoroadenine-9-β-D-arabinofuranoside	Antineoplastic	42.24	1. Antineoplastic [25] 2. Immunosuppressant [26]
4.	Retinoic acid p-hydroxyanilide	Antineoplastic	2.38	1. Antineoplastic [27]
5.	Docetaxel	Antineoplastic	2.26	1. Antineoplastic [28,29] 2. Antiviral-EBV [30]
6.	Evodiamine	Antineoplastic	10.24	1. Antineoplastic [31,32] 2. Anti-inflammatory [33] 3. Antiviral [34,35,36], viral, and bacterial pneumonia [37]
7.	Staurosporine from Streptomyces sp.	Antibiotic	0.08	1. Antineoplastic [38,39] 2. Antimicrobial [40]
8.	Resveratrol	Anti-inflammatory	40.82	1. Antilipemic and antidiabetic [41] 2. Anti-inflammatory [42,43] 3. Antineoplastic [44] 4. Alzheimer’s disease pathomechanism modulator [45] 5. Antioxidant and antimicrobial [46] 6. Antiviral [47] 7. Cardioprotective [48]
9.	Metyrapone	Adrenal steroid synthesis inhibitor	19,554	1. 11β-hydroxylase enzyme inhibitor [49] 2. Antidepressant [50]
10.	(+)-JQ1	Bromodomain inhibitor (Thienotriazolodiazepine)	0.82	1. Bromodomain inhibitor [51] 2. Reduces IFN-γ expression [52] 3. Antineoplastic [53,54] 4. Cardioprotective [55,56] 5. Anti-inflammatory [57]
11.	Givinostat hydrochloride hydrate	Hydroxamate inhibitor	1.61	1. Anti-inflammatory [58] 2. Cardioprotective [59] 3. Antineoplastic/antiangiogenic [60]
12.	Thapsigargin	Calcium channel blocker	1.38	1. Antineoplastic [61] 2. Endoplasmic reticulum Ca ^2+^ inhibitor [62] 3. Antiviral [63,64]
13.	Enalaprilat	Angiotensin-converting enzyme (ACE) inhibitor	80.83	1. Treatment of hypertension and hypertensive heart failure [65]
14.	5α-Androstan-3β-ol	Steroid	23.13	No reported activity
15.	Elvitegravir	Antiviral	12.7	1. Antiretroviral [66,67]
16.	Lomibuvir	Antiviral	38.54	1. Anti-HCV activity [68,69] 2. SARS-CoV-2 [70]

**Table 2 viruses-14-02150-t002:** Summary of effective inhibition under different treatment conditions.

Sr. No.	Compound Name	Chemical Structure	CC50 (µM)	Maximum Concentration (µM)	Log Difference Effectiveness against DENV-2	EC50 (µM)	Selectivity Index (SI)
1	Doxorubicin	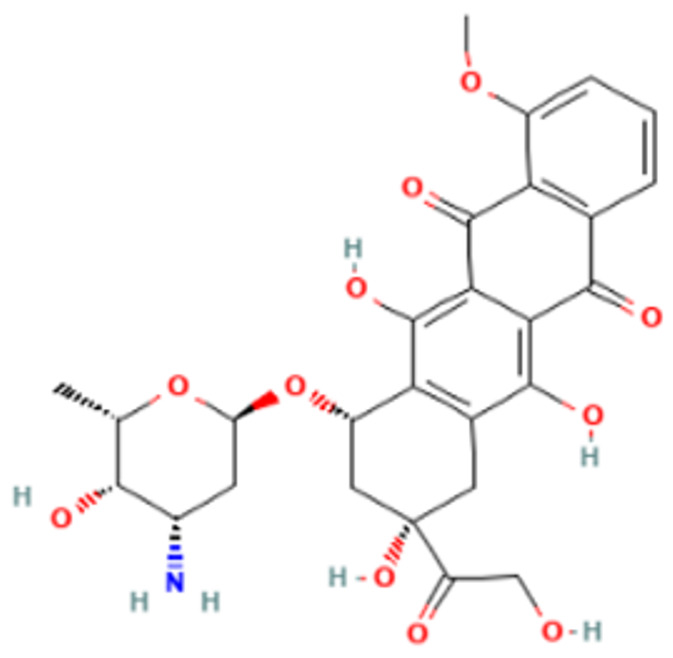	116.9	25	Post-treatment—1.453 Co-treatment—4.958	19.99 6.573	5.848 17.785
				12.5	Co-treatment—4.377	6.573	17.785
				6.25	Co-treatment—2.107	6.573	16.115
2	Resveratrol	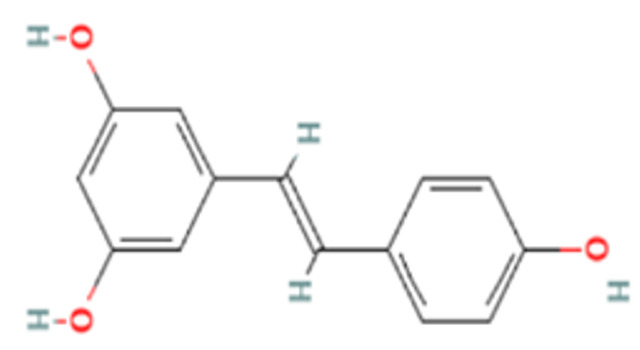	40.54	12.5	Post-treatment—1.226 Pre-treatment—1.188	4.013 7.592	10.102 5.340
				6.25	Post-treatment—1.115	4.013	10.102
3	Enalaprilat	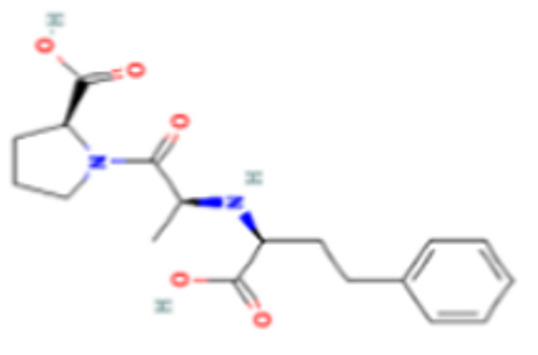	80.83	1.56	Post-treatment—1.239	1.079	74.911
4	Elvitegravir	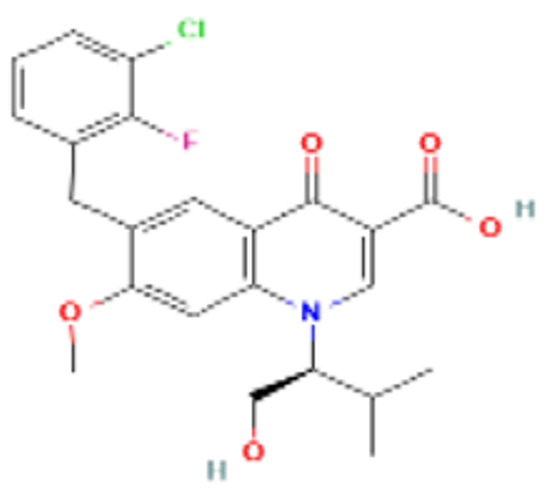	12.7	6.25	Pre-treatment—1.316	4.405	2.883
5	Lomibuvir	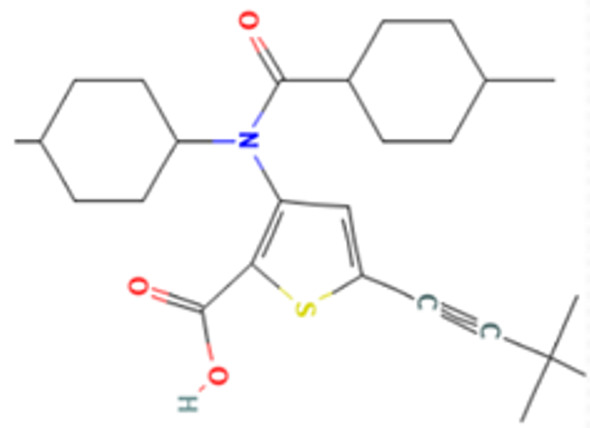	38.54	6.25	Pre-treatment—1.224	3.740	10.305

## Data Availability

Not applicable.

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
