# Peer review of "A Transcriptomics-Based Bioinformatics Approach for Identification and In Vitro Screening of FDA-Approved Drugs for Repurposing against Dengue Virus-2"

_viruses, 2022, doi:10.3390/v14102150_

Round 1
Reviewer 1 Report
Punekar M et al reported a transcriptomics-based bioinformatics approach for identifi- 2 cation and in-vitro screening of FDA-approved drugs for repur- 3 posing against dengue virus-2. Firsly, identification of differential gene expression between dengue patients and healthy individuals were performed using computation analysis form three microarray gene expression datasets. Secondly, identification of drug candidates was subsequently done using CMap. Thirdly, the compounds, which demonstrated antiviral activity were evaluated for anti-viral activity at different concentrations under pre-treatment, co-treatment, and post-treatment. Finally, molecular docking studies with DENV protein targets were further used to identify potential compound-viral protein interactions.
The manuscript is well-written with support evidence. However, there are some concerns that need to be clarified before publication. The methods used in antiviral screening of drugs may need modifications. In this study, Vero cells were treated with non-toxic dose of drugs for 24 hours in a pre, co and post-infection and then incubated for 120 hours with DENV. Totally 120 hours seem to be too long for the incubation time with DENV. DENV-infected cell lines generally induced apoptosis at 48 hours post infection. Therefore, the reduction of viral titer may come from the reduced number of cells. It will be good if the cell viability in DENV-infected cells with selected drug will be shown. Also, the MOI that were used in this study should be documented. In addition, experiments were performed in triplicates in two independent trials. At least three independent experiments should be performed before stastistical analysis.
Author Response
Independent Review Report, Reviewer 1
The manuscript is well-written with support evidence. However, there are some concerns that need to be clarified before publication. The methods used in antiviral screening of drugs may need modifications.
Vero cells were treated with non-toxic dose of drugs for 24 hours in a pre, co and post-infection and then incubated for 120 hours with DENV. Totally 120 hours seem to be too long for the incubation time with DENV. DENV-infected cell lines generally induced apoptosis at 48 hours post infection. Therefore, the reduction of viral titer may come from the reduced number of cells. It will be good if the cell viability in DENV-infected cells with selected drug will be shown
Response: We thank the reviewer for his and critical comments which have improved the manuscript. Based on our standardization experiments, we have observed that DENV-2 production in Vero-CCL-81 was maximum on 5th day. Hence, we have used 120 hrs for our studies.
As suggested, we have performed additional experiments to investigate the effect of the drugs on viability in infected cells treatment under different conditions. The experiments were conducted at both 3 days and 5 days. The results revealed that percent cell viability is greater in infected cell cultures that received treatment compared to infected cells which did not receive treatment (Virus control) irrespective of days. This suggests that drugs by inhibiting DENV replication increases the cell viability and reduction in viral titre is not due to the cell death. The results have been provided in the revised version (Section 3.7; lines 346-359) and figure 7.
Also, the MOI that were used in this study should be documented.
Response: We have used an MOI of 0.1 which is mentioned in the revised version (line 129)
In addition, experiments were performed in triplicates in two independent trials. At least three independent experiments should be performed before statistical analysis.
Response: As suggested, we performed an additional set of detailed study experiments and the results of triplicates performed in independent points have been provided We have modified the graphs and data by including the results of all the experiments (Lines 281-305).

Reviewer 2 Report
In this study, Madhura Punekar employed a transcriptomics-based bioinformatics approach for drug identification against dengue virus. Focus forming unit assay and quantitative RT-PCR were used to evaluate the antiviral activity of those drug. The targets of those drugs were also discussed by molecular docking studies. This research would facilitate clinical trials work.
1. Antiviral assay, Focus Forming Unit assay and Real-time RT-PCR were mentioned too briefly and should be described in detail in Materials and Methods.
2. References should be noted in line 357.
3. The references are incomplete in discussion.(BMC Res Notes . 2018 May 16;11(1):307. doi: 10.1186/s13104-018-3417-3.Screening of melatonin, α-tocopherol, folic acid, acetyl-L-carnitine and resveratrol for anti-dengue 2 virus activity)
Author Response
- Antiviral assay, Focus Forming Unit assay and Real-time RT-PCR were mentioned too briefly and should be described in detail in Materials and Methods.
Response: Thank you for indicating mistakes, modified accordingly in sections 2.6 and 2.7 (lines 126-173)
- References should be noted in line 357.
Response: Thank you for indicating mistakes, references have been added and can be noted on line number 421
- The references are incomplete in discussion.(BMC Res Notes . 2018 May 16;11(1):307. doi: 10.1186/s13104-018-3417-3.Screening of melatonin, α-tocopherol, folic acid, acetyl-L-carnitine and resveratrol for anti-dengue 2 virus activity)
Response: Thank you for suggesting, a reference has been cited (line no 438; Ref no 41) and can be found on line number 596.

Round 2
Reviewer 1 Report
The authors addressed the concerns in the revised manuscript.
Author Response
The authors addressed the concerns in the revised manuscript.
(x) English language and style are fine/minor spell check required
Response: We thank the reviewer for his time and his suggestions. We have checked the manuscripts for typos and grammatical errors and have corrected the same in the revised manuscript.